# Get Spliced: Uniting Alternative Splicing and Arthritis

**DOI:** 10.3390/ijms25158123

**Published:** 2024-07-25

**Authors:** Maurice J. H. van Haaren, Levina Bertina Steller, Sebastiaan J. Vastert, Jorg J. A. Calis, Jorg van Loosdregt

**Affiliations:** 1Center for Translational Immunology, University Medical Center Utrecht, 3584 CX Utrecht, The Netherlands; 2Division of Pediatric Rheumatology and Immunology, Wilhelmina Children’s Hospital, 3584 CX Utrecht, The Netherlands

**Keywords:** alternative splicing, arthritis, immunology

## Abstract

Immune responses demand the rapid and precise regulation of gene protein expression. Splicing is a crucial step in this process; ~95% of protein-coding gene transcripts are spliced during mRNA maturation. Alternative splicing allows for distinct functional regulation, as it can affect transcript degradation and can lead to alternative functional protein isoforms. There is increasing evidence that splicing can directly regulate immune responses. For several genes, immune cells display dramatic changes in isoform-level transcript expression patterns upon activation. Recent advances in long-read RNA sequencing assays have enabled an unbiased and complete description of transcript isoform expression patterns. With an increasing amount of cell types and conditions that have been analyzed with such assays, thousands of novel transcript isoforms have been identified. Alternative splicing has been associated with autoimmune diseases, including arthritis. Here, GWASs revealed that SNPs associated with arthritis are enriched in splice sites. In this review, we will discuss how alternative splicing is involved in immune responses and how the dysregulation of alternative splicing can contribute to arthritis pathogenesis. In addition, we will discuss the therapeutic potential of modulating alternative splicing, which includes examples of spliceform-based biomarkers for disease severity or disease subtype, splicing manipulation using antisense oligonucleotides, and the targeting of specific immune-related spliceforms using antibodies.

## 1. Understanding Alternative Splicing: Principles, Regulation, and Outcomes

Regulating gene expression is important for numerous cellular processes, and vital for cell survival and overall activity. One important mechanism to regulate expression is by splicing. For the majority of mRNA transcripts, splicing occurs constitutively at standard splice sites, meaning that introns are removed and exons are joined together in their genomic order [1]. In addition to constitutive splicing, alternative splicing (AS) can occur (Figure 1). It is estimated that nearly 95% of genes with multiple exons undergo some form of AS [2]. AS serves two primary functional purposes: first, it provides an additional layer of gene expression control, permitting the precise adjustment of protein levels by regulating the degradation of mRNA, and second, AS can give rise to functionally distinct proteins when alternative mRNA transcripts are translated. AS can be classified into five types (Figure 1). (1) Intron retention leading to the inclusion of intron(s) in the mature mRNA, often preventing translation through nonsense-mediated decay (NMD) and sometimes leading to pre-terminal stop codons yielding truncated products. (2) Mutually exclusive exons leading to mRNA containing either one of two exons but never both simultaneously. (3) Cassette alternative exons, where one exon is spliced out or retained, leading to two different isoforms. (4) Alternative 5′ and (5) 3′ splice sites resulting in either shorter or longer exons, with the 5′ and 3′ giving information about the location of the difference. These forms of alternative splicing have the potential to generate a vast array of protein isoforms from a single gene, each with potential biological and developmental implications due to variations in protein properties such as localization signals, binding capabilities, and enzymatic activity. The precise regulation of AS is crucial for maintaining an accurate expression of mRNA and protein isoforms, ultimately contributing to normal cellular function. Variations in alternative splicing (AS) patterns can manifest in response to environmental and developmental cues, exerting significant influence over biological and developmental processes [3]. AS is controlled by a complex and still incompletely understood process involving many regulatory proteins that together form the spliceosome. The details regarding this process are outside the scope of this manuscript and have been reviewed in detail elsewhere [4]. Although the majority of eukaryotic genes undergo AS, the exact molecular mechanisms underlying the AS of individual mRNAs, and the extent to which AS regulation contributes to cellular and developmental processes in different cell types remain to be further elucidated. Splicing mainly takes place co-transcriptionally and several transcriptional mechanisms of AS regulation are identified. In general, several splicing factors, including serine/arginine-rich splicing factors (SRSF) and heterogeneous nuclear ribonucleoproteins (hnRNPs), and other RNA-binding proteins (RBPs), are present on a single mRNA simultaneously, and ultimately, the balance between splicing enhancing and repressing factors determines AS patterns. It is believed that many AS events are currently unknown; however, novel technologies and bioinformatic methodologies can aid in the description of the full isoform landscape. The recent rise of third-generation RNA sequencing technologies such as PacBio single molecule real-time (SMRT) and Oxford Nanopore Technologies (ONT) sequencing (Box 1) have allowed for a more inclusive and rapid identification of novel transcript isoforms, shedding light on the intricacies of AS within these contexts.

Box 1Determining transcript isoforms using third generation RNA sequencing techniques.  Third-generation sequencing techniques, such as PacBio single molecule, real-time (SMRT) and Oxford Nanopore Technologies (ONT) sequencing, have revolutionized the field of genomics by enabling high-throughput sequencing of individual native DNA and RNA molecules. In PacBio SMRT sequencing, sequencing units are contained within specialized chambers known as zero-mode waveguides, where fluorescent light is emitted and detected as nucleotides are incorporated into the growing sequences. This process allows for the synthesis-based sequencing of transcripts, with read lengths reaching up to 65 kilobases. Nanopore sequencing utilizes a pore, through which electrical currents are monitored. The passage of nucleotides through the nanopore modulates the electrical signal, and this signal is translated into the corresponding DNA or RNA sequence. Notably, nanopore sequencing is not constrained by read length limitations, making it particularly suitable for obtaining full-length transcript sequences. This characteristic enables the straightforward elucidation of splice patterns, facilitating the generation of isoform-level transcriptome profiles.

### Alternative Splicing as a Post-Transcriptional Regulator of Dynamic Cell Responses

As previously mentioned, alternative splicing is a complexly regulated mechanism allowing for rapid cellular adaptation. There are currently many unidentified isoforms, both in normal physiology and disease states, that convolute our understanding of many pathways, including (but not limited to) immune regulatory pathways. By extending our knowledge regarding these unidentified isoforms, we can increase our understanding of diverse pathways. We will further describe how AS is involved in immune activation, with a focus on how AS is (dys)regulated in immune-mediated arthritis. We will also describe future perspectives for elucidating isoform functionality and potential therapeutic possibilities.

## 2. Alternative Splicing Plays an Important Role in the Immune Response

It is increasingly evident that AS plays a pivotal role in complex cellular systems that necessitate rapid and dynamic alterations in gene expression and function in response to developmental and environmental cues [5]. The immune system serves as a prime example of such a complex system. Here, the detection of pathogens initiates signaling cascades that lead to swift and precise changes in gene expression, crucial for the defense against invading pathogens. AS has been described to be involved in these rapid changes in protein expression and diversity [6]. Despite the growing recognition of AS as a regulator of gene expression during immune responses, the precise molecular mechanisms governing AS in various immune cell types and its regulation throughout different stages of the immune response remain largely unknown. Immune cells can be classified into two major categories: Firstly, innate immune cells, which are responsible for the initial, non-specific defense against invading pathogens, such as monocytes, macrophages, natural killer (NK) cells, and dendritic cells (DCs). The second major category of immune cells are those involved in the adaptive immune response that targets specific antigens, encompassing B and T lymphocytes [7]. Advances in RNA sequencing techniques have allowed for the identification and quantification of AS in various cell types during infections [5,8,9,10]. In this context, several instances of alterations in AS in response to infection are described, along with their functional implications in immune cells [11]. RNA-seq has been used to detect thousands of isoform differences upon activation in various immune cell lineages [12]. Here, AS was demonstrated to play a pivotal role in the maturation and differentiation of hematopoietic precursor cells into fully functional immune cells [12]. One such example is the extensively studied CD45 transcript. CD45 is a transmembrane glycoprotein found on T and B cells. The CD45 transcript can be spliced in eight possible mRNA isoforms, five of which have been observed in humans [13,14,15]. The variation in exon inclusion influences TCR signaling and cytokine production, and is also used to identify B and T cell subsets. B cells have exons 4, 5, and 6, naïve T cells have either 4 or 5, and memory T cells lack all three exons. In cases of aberrant splicing due to a C77G polymorphism in exon 4, splicing is prevented and associated with impaired T cell function and with an increased susceptibility to human immunodeficiency virus infection. This leads to an increased CD45RA, which is typically present on naïve and central memory T cells. The specific regulation of the splicing of CD45 is realized by an interplay of multiple factors and regulators and underscores the intricate nature of changes in splicing patterns during the maturation and differentiation of immune cells. While much remains unknown, there are specific examples in the innate immune system that have been thoroughly elucidated.

### 2.1. Alternative Splicing as a Regulator of Cytokine Expression in Innate Immune Cells

AS has been demonstrated to be involved in cytokine expression, for instance for IL-6 regulation (Figure 2A) [16]. IL-6 is a central player in host defense and performs various functions, including triggering the acute phase response [17]. In quiescent macrophages, heterogeneous nuclear ribonucleoprotein M (hnRNP M) is poised on chromatin in an inactive state, preventing the maturation of IL-6 transcripts by blocking the removal of introns. Upon even the slightest detection of a pathogen, the nuclear factor kappa-light-chain-enhancer of B cells (NFκB) initiates the transcription of various pro-inflammatory genes, including *IL6*. This will only directly result in a marginal increased expression of IL-6 if hnRNP M is still preventing intron removal. Only upon full activation is the macrophage hnRNP M phosphorylated. Subsequently, the removal of introns is allowed, and mature IL-6 mRNA levels quickly rise for translation. Here, hnRNP M acts as a safeguard, restraining the initial activation of the innate immune response until the pathogen has been fully processed [16].

### 2.2. Alternative Splicing as an Inflammasome Activation Regulator

Additionally, AS has been demonstrated to be involved in inflammasome activation, and more specifically in the NLR family pyrin domain-containing 3 (NLRP3) protein (Figure 2B). NLRP3 is a component of the NLRP3 inflammasome and functions as a pattern recognition receptor. Cell activation results in a switch in AS where exon 5 is included, leading to oligomerization and allowing for the recruitment of apoptosis-associated speck-like proteins (ASC) and pro-caspase-1. Consequently, this active inflammasome complex allows caspase-1 to become activated, thereby cleaving pro-Interleukin-1β (IL-1β) and pro-Interleukin-18 (IL-18) into their active forms. Notably, macrophages have been shown to stochastically transcribe the shorter isoform of NLRP3, which excludes exon 5 [18]. This suggests that AS of NLRP3 may serve as a mechanism for fine-tuning the response of macrophages to pathogenic invasions and regulating caspase-1-mediated programmed cell death in macrophages.

### 2.3. Alternative Splicing as a Regulator of Innate Immune Cell Activation

AS has also been demonstrated to be involved in the activation of innate immune cells through the regulation of the glycoprotein tenascin-c (TNC, Figure 2C) [19]. TNC is a component of the extracellular matrix, and involved in tissue repair, cellular stress, and immune activation. In non-activated DCs and quiescent macrophages, there is a low expression of large TNC isoforms that include the so-called AD2AD1 domains. This protein appears in fibrillar structures and promotes adhesion to fibroblasts and prevents the chemotaxis of DCs and cytokine synthesis by macrophages. Upon activation, IL1β increases the expression of small tenascin-C isoforms, which lack the AD2AD1 domains. This isoform is a secreted soluble form, thereby preventing adhesion to fibroblasts, and consequently DCs can infiltrate the infected tissue and macrophages become activated. The increased expression of these small isoforms is also observed in patients with rheumatoid arthritis (RA) [19]. These examples of AS in innate immune cells demonstrate that AS is required for an efficient innate immune response.

### 2.4. Alternative Splicing as a Regulator of Cytokine Expression in Adaptive Immune Cells

Having discussed some representative examples of AS in innate immune cells, there are also various studies that demonstrate that AS helps to modulate the adaptive immune system [13,15,20,21]. For instance, AS has been demonstrated to be involved in cytokine expression by regulating a member of the SLAM family (SLAMF, Figure 2D) of immune receptors [22]. SLAM receptors are receptors that are capable of activating various downstream pathways. The canonical SLAMF6 transcript is constitutively expressed in T cells. It is hypothesized that this isoform inhibits T cell function by recruiting SLAM associated protein (SAP), thereby effectively preventing cytokine production [22]. In contrast, an alternative SLAM6 isoform that is lacking exon 2 (SLAMF6Δ2) has a co-stimulatory effect on T cell function and leads to increased cytokine expression [22]. This isoform is similar to SLAMF6-FL, regarding transmembrane and intracellular domains, but one major difference is that the immunoglobulin variable is completely absent. Additionally, SLAMF6Δ2 has increased self-binding compared to its SLAMF6-FL counterpart, allowing for the inhibition of SLAMF6-FL. The inhibitory canonical isoform is more prevalent in exhausted T cells that have lost their effector function, such as during chronic inflammation or in some cases of cancer, whereas the shorter isoform is more abundantly present in activated T cells. Given that the splicing ratio of SLAMF6 determines T cell function, manipulating this ratio between the shorter and full-length isoform may be a promising therapeutic approach for treating patients with chronic inflammation or cancer [22].

### 2.5. Alternative Splicing as a Regulator of T Cell Subset Differentiation

AS has been implicated in CD4+ T cell subset differentiation by regulating forkhead box P3 (FOXP3, Figure 2E), which is an essential transcription factor for the function and differentiation of regulatory T (Treg) cells. In unstimulated conditions, full-length FOXP3 stimulates regulatory T cell development and prevents Th17 differentiation through RORC2 inhibition [23,24,25,26]. Upon stimulation and in the presence of IL-1β, exons 2 and 7 of FOXP3 transcripts are spliced out, resulting in a dominant negative isoform. The reason for this is that (1) exon 2 contains an LxxLL motif that interacts with RORA and RORC2 and (2) Exon 7 encodes a leucine zipper domain which is required for FOXP3 dimerization and Treg function [27]. Therefore, T cells are directed toward Th17 differentiation, resulting in IL-2 and IL17A production. The exact mechanisms that regulate the AS of FOXP3 remain unknown.

### 2.6. Alternative Splicing as a Regulator of NFκB Signalling in Adaptive Immune Cells

One final example demonstrates that AS is involved in NFκB signaling by regulating TNF-receptor associated factor 3 (TRAF3, Figure 2F). TRAF3 is an adapter protein and one of the components of the NFκB-inducing kinase complex, together with TRAF2, NFκB-inducing kinase (NIK), and cIAP [28]. In quiescent conditions, NIK binds to TRAF3 which is bound to TRAF2 and cIAP. cIAP is ubiquitinated and therefore the entire complex, including NIK, is degraded. However, upon activation, exon 8 is spliced out. TRAF3Δ8 is unable to bind NIK, and therefore NIK is not degraded with the complex. Then, NIK initiates the non-canonical NFκB pathway by activating IKK-α, which in turn phosphorylates NFKB2, ultimately resulting in the increased expressions of CxCL13, CCL21, and CCL19 [29,30]. In summary, the examples described above underscore the role of AS in the regulation of immune responses during infections. There are only a few detailed examples of AS in the regulation of adaptive immune cells, and much remains to be elucidated. Ideally, a comprehensive atlas should be created of all isoforms in various immune cell subtypes that are exposed to different stimuli. This will provide information about the way that isoforms influence proteins or pathways. This atlas should be extended to include cells from patients with immune-related diseases. Further investigation is necessary to elucidate the mechanisms by which immune activation regulates splicing and the way that alternative splicing is involved in immune-related diseases. In turn, this information can be used to detect biomarkers for immune activation or disease and even assist in designing therapeutic options [31].

## 3. Alternative Splicing in the Scope of Inflammatory Arthritis

AS has been demonstrated to play a significant role in immune cells and has been associated with immune-mediated diseases. Genetic associations have been made between AS and several immune-related diseases, including rheumatoid arthritis (RA). Genome-wide association studies (GWASs) have identified more than 100 genetic loci that link with RA [32]. Whether specific genomic variations could affect splicing can be assessed by splicing quantitative trait loci (sQTLs). sQTLs refer to loci associated with RNA splicing variations and affect the splicing of pre-mRNA, leading to differences in the expression levels of different mRNA isoforms. sQTLs are often located in introns and frequently target global splicing patterns of genes, instead of individual splicing events. sQTLs affect the binding of RBPs by modifying RBP binding sites, thereby altering the splice site strength. There is a significant enrichment of single-nucleotide polymorphisms in sQTLs among disease-associated loci when using the genome-wide association studies data [33]. The top 10 diseases with the largest numbers of genome-wide significantly associated SNPs include multiple sclerosis, psoriasis, and rheumatoid arthritis. Here, a significant enrichment of sQTL SNPs was found among the loci associated with rheumatoid arthritis when excluding MHC (*p* = 0.032, Odds ratio (OR) around 2 when comparing sQTL with non-sQTL, one-tailed Fisher’s exact test with Bonferroni correction) and on the border of significance when including MHC (*p* = 0.064, OR around 1.8). One study examined sQTLs across multiple human tissues and compared this to several diseases [34]. When compared to non-sQTLs, sQTLs displayed a substantial enrichment in variants associated with a wide variety traits and diseases. AS was demonstrated to significantly link immune-related diseases, encompassing several autoimmune disorders and chronic inflammatory conditions such as multiple sclerosis (MS), inflammatory bowel disease (IBD), and arthritis [35,36,37,38,39]. Taken together, these studies indicate that the dysregulation of alternative splicing can contribute to the pathogenesis of immune-related diseases such as rheumatoid arthritis. Gaining a deeper understanding of the specific changes in alternative splicing patterns associated with various diseases could potentially help to understand disease pathogenesis, improve diagnostics, and revolutionize therapeutic strategies [40].

In this review, we focus on inflammatory arthritis, a complex inflammatory disease characterized by the inflammation of one or several joints leading to joint damage and accompanied by pain. Rheumatoid arthritis is the most common form of inflammatory arthritis, with a yearly incidence of 2–5 per 10,000 population in Europe [41]. There are different examples of AS involvement in arthritis pathogenesis, but while significant advances have been made, our understanding of cellular and molecular mechanisms remains limited [32,42].

### 3.1. Alternative Splicing of CD44 in Arthritis

One of the studies that have assessed AS in arthritis focuses on CD44. CD44 is a cell-surface receptor with at least two known ligands, hyaluronic acid and galectin 8 (Figure 3A) [43]. While CD44 is commonly recognized as an important regulator in cancer biology, recent findings indicate its relevance in immunology, where it has many roles, which include regulating apoptosis and inflammation [44,45,46]. CD44 comprises 10 constitutive exons and 10 variable exons. CD44 exhibits significant diversity, with more than 20 isoforms described due to the inclusion or exclusion of the 10 variable exons in various combinations [44,45,47]. In the context of non-arthritic inflammation, the interaction between the full-length CD44 and galectin-8 leads to the induction of apoptosis. However, soluble isoforms, known as sCD44v5 and sCD44v6, exist and have been demonstrated to be expressed in the serum of patients with rheumatoid arthritis. Here, they display elevated expressions compared to both the healthy control and miscellaneous inflammatory rheumatic diseases (MIRD). However, the cellular origin of sCD44 variant isoforms is currently unknown. The presence of sCD44v5 and sCD44v6 correlates with disease severity [48,49]. These isoforms are a product of AS and proteolytic shedding, where the isoforms lose their transmembrane domains. These soluble CD44 isoforms can capture galectin-8 without initiating downstream intracellular signaling pathways. As a result, apoptosis is not induced, and inflammation persists. The use of anti-CD44 monoclonal antibodies has shown a reduction in inflammation in arthritic mice [49]. In summary, the abnormal alternative splicing (AS) of transcripts that encode transmembrane cytokine receptors, in this case CD44, may lead to an imbalance in the expression ratio between surface-bound and soluble isoforms. Similar receptors could be targeted to mitigate pro-inflammatory signaling.

### 3.2. Alternative Splicing of IL-6R in Arthritis

One example of targeting similar receptors to CD44 is the following case. In this case, rheumatoid arthritis (RA) patients are treated with Tocilizumab, a competitive antagonist that targets the interaction between IL-6 and a soluble variant of its receptor IL-6R (sIL-6R, Figure 3B). By blocking the IL-6 interaction with sIL-6R, it prevents a strong pro-inflammatory gp130-mediated signal transduction cascade that can evoke a cytokine storm. sIL-6R is generated from two sources, firstly alternative splicing, which leads to the excision of exon 9, encoding the transmembrane region, and secondly proteases, which can also cleave membrane-bound IL-6R through proteolytic cleaving [26,50]. IL-6 and sIL-6R are both elevated in RA, and the targeting of IL-6 and sIL-6R results in a significant reduction in disease symptoms [50,51,52]. Manipulating the AS of IL-6R to impair the expression of sIL-6R might be worth exploring to reduce the disease burden in these patients.

### 3.3. Alternative Splicing of Survivin in Arthritis

Another case in which AS is involved in RA pertains to survivin (Figure 3C). Survivin is a member of the inhibitor of apoptosis (IAP) family and is encoded by the *BIRC5* gene [53,54]. Survivin has a multitude of functions which depend on both the location and the isoform expressed. The general notion is that survivin is essential for regulating cell division, inhibiting apoptosis and tissue repair. However, the function of survivin differs according to location. Both cytoplasmic and mitochondrial survivin promote cell proliferation and inhibit apoptosis, while nuclear survivin plays a regulatory role in cell division [53]. Survivin can be exported to the cytoplasm from the nucleus to promote anti-apoptotic functions by forming a complex with X-linked IAPs (XIAP). This complex binds and inhibits caspase-3 and -9, thereby effectively inhibiting apoptosis. Mitochondrial survivin binds to pro-apoptotic protein Smac/Diablo, thereby inhibiting the release of Smac/Diablo and thus preventing the activation of caspase-9. In addition to the difference in function based on cellular location, there are also different isoforms exerting different functions. Six different isoforms exist, of which three are the most frequent: full-length survivin (FL), survivin including an additional exon 2 insert (2B), and survivin without exon 3 (Δ3) [53]. While FL and 2B can be actively exported from the nucleus, Δ3 resides in the nucleus as it lacks an export signal. The Δ3 isoform is a dual functional protein that shares anti-apoptotic functions with FL and additionally has roles for cell migration and cell stability. The 2b and Δ3 survivin isoforms can dimerize, forming either homodimers or heterodimers. The function of survivin is determined by the specific dimers that are formed. One study found that CD19^+^ B cells from the peripheral blood of RA patients displayed a high production of 2B and Δ3 compared to healthy controls, whereas the FL levels were similar [53]. These data suggest that it is not the quantity of the individual splice variants but instead the proportional composition between the splice variants that is of clinical relevance in RA. Typically, an excess of FL with low 2B/FL or low Δ3/FL complexes identified patients with increased disease activity. By therapeutically depleting B cells using rituximab, a splice shift can occur, leading to reduced FL and increased 2B and Δ3, which could result in reduced disease activity [53]. While the limited examples above give some insight into the relevance of alternative splicing in the context of arthritis, there is much that remains unknown. It is prudent to compile an inventory of established cases. For this reason, a comprehensive summary (Figure 4) was composed to display the current understanding of alternative splicing in the context of arthritis [2,6,9,10,12,17,21,24,27,30,32,33,34,36,37,38,39,41,42,46,50,51,52,53,54,55,56,57,58,59,60,61,62,63,64,65,66,67,68,69].

## 4. The Clinical Relevance of Alternative Splicing in Arthritis

Here, we outlined that AS is implicated in immune regulation and in inflammatory arthritis. Mapping changes in how alternative splicing impacts isoform expression and understanding the consequences for immune activation and immune-related diseases could have different clinical implications (Figure 5). More specifically, AS events can be further explored for their therapeutic potential in the development of diagnostic and prognostic biomarkers. Potentially, specific AS events can be targeted with diverse strategies to suppress disease activity. Here, we provide three examples of such strategies.

### 4.1. Antisense Oligonucleotide Therapy for the Suppression of Proteins

Firstly, antisense oligonucleotide (ASO, Figure 5A) therapy can be used to manipulate isoform expression. Here, antisense oligonucleotides can be designed to bind specific splice sites, thereby preventing the binding of the spliceosome and thus skewing splicing towards a specific isoform. This strategy was proved effective in a study that targeted the Tumor Necrosis Factor (TNF) signaling pathway in collagen-induced arthritic (CIA) mouse models [59]. As TNF signaling is directly involved in inflammation, it is interesting to target this pathway. ASOs were utilized to prevent the inclusion of exon 7 in TFN receptor TNFR2. This exon 7 contains the transmembrane domain and therefore TNFR2Δ7 is excreted and capable of capturing circulating TNF-α. Concurrently, the expression of the functional membrane-bound isoform is also reduced, thus offering a two-pronged method to control inflammation. After five days of treatment, the mice displayed a 40% increased survival rate compared to untreated controls. Additionally, these mice displayed significantly reduced paw swelling and reduced clinical scores compared to saline controls. The TNFR2Δ7 protein was actively present in the serum until at least day 50, which was 30 days after the final ASO injection, indicating that it is a stable method for targeting the TNF signaling pathway.

### 4.2. Isoform-Specific Targeting with Monoclonal Antibodies

Secondly, isoform-specific monoclonal antibodies (mAb) can be utilized to detect and bind proteins harboring exons that increase the pro-inflammatory function of specific proteins. One study employed this strategy to target specific adiponectin isoforms and demonstrate their potential for rheumatoid arthritis treatment (Figure 5B). Adiponectin is a hormone protein implicated in the regulation of glucose levels, fatty acid breakdown, and also inflammation [57]. There are three main oligomeric forms, of which two isoforms can increase the expression of chemokines and pro-inflammatory cytokines [70]. Utilizing isoform-specific adiponectin mAbs, a moderate inhibition of the expression of pro-inflammatory cytokines, such as IL-6 and IL-8, was observed in vitro in human cells [70]. Similarly, in a CIA mouse model, injection with these mAbs decreased TNF-α and IL-6 levels; additionally, paw volume and squeaking were reduced, and the anti-arthritic effects were also histologically confirmed.

### 4.3. Utilizing Antagonistic Isoforms to Counter Inflammation

The final therapeutic strategy utilizes naturally existing soluble receptor isoforms (Figure 5C) that are antagonists of cytokine signaling. As recombinant isoforms, they can be administered to disrupt specific pro-inflammatory cytokine/receptor complexes, thereby suppressing cytokine effects. An exemplary case is glycoprotein 130 (gp130) [71]. Gp130 is a member of the IL-6 receptor complex, and has been implicated in many processes such as hematopoiesis, immune response, and inflammation [67]. A recombinant protein was generated from the naturally occurring soluble human gp130 isoform. This soluble isoform acts as an antagonist of IL-6 receptor signaling. In this study, the immunohistochemical analysis of phosphorylated STAT-3 in the synovial tissue of AIA mice demonstrated that soluble gp130 has a concentration-dependent inhibitory effect. This effect was confirmed in vitro in human synovial fibroblasts stimulated with IL-6. Additionally, using soluble gp130 in an experimental murine model of acute peritonitis led to the reduced expression of CCL5, and reduced immune cell recruitment.

### 4.4. Challenges, Limitations, and Future Prospects

The previously mentioned therapeutic examples are just a few of the recent cases that have highlighted the potential of targeting AS and aberrant isoforms in arthritis. It is important to note that there are still technical difficulties to overcome. Some examples of these difficulties include the following: (1) Due to their relatively short length of 15 to 25 nucleotides, ASOs may display some off-target effects in vivo that are difficult to predict [64]. The treatment options must be tested extensively to detect and eliminate the chance of potential off-target effects. One study, using gapmers, short DNA antisense oligonucleotides with RNA-like segments on both sides, showed reduced off-target effects [69]. This study states that this is due to the extension of oligonucleotides. Another study states that the delivery of ASOs can be enhanced by using endosomolytic compounds [55]. These compounds can be used to release ASOs that otherwise accumulate in endosomes. In a similar manner, as there are off-target effects using ASOs, antibodies and antagonistic proteins might also interact with unexpected targets. (2) The in vivo delivery method should be considered thoroughly if these strategies are applied during the treatment of human diseases. These methods listed above have primarily been tested in mouse models or cell lines and might not accurately represent human in vivo conditions, despite biochemical similarities. There might be a difference in stability or efficacy. To conclude, while it is clear that the aforementioned strategies have incredible potential as a therapeutic strategy for immune-related diseases, such as inflammatory arthritis, there are still challenges to overcome that hopefully will be addressed in the near future.

## 5. Conclusions

In this review, we described alternative splicing as a complex post-transcriptional process that is directly implicated in various immunological processes in both the innate and adaptive immune systems. In addition, we discussed AS in the context of autoimmune diseases, specifically in arthritis, and assessed the therapeutic potential of AS manipulation for immune-mediated diseases. Currently there are a small number of splicing modifiers that have been approved by the FDA for treatment of spinal muscular atrophy (SMA) and Duchenne muscular dystrophy (DMD) [65]. These RNA-targeting splicing modifiers open up a plethora of novel treatment options for many diseases, including immune-mediated diseases. In immune cells, the precise functions of many alternatively spliced transcripts still need to be fully understood. Ideally, the third-generation long-read RNA sequencing of specific cell-types in patients and healthy donors may yield valuable information to further elucidate pathways that are currently not well understood. Presently, several efforts are being made to generate a so-called atlas of all isoforms that are expressed in different human immune cells using long-read RNA sequencing [9,10,11,62,66,68]. When AS is better understood in immune activation and immune-mediated diseases, RNA-targeting splicing modifiers can be explored for their potential therapeutic value. In addition, future research focused on understanding how specific single-nucleotide polymorphisms (SNPs) that modulate protein isoform expression can contribute to susceptibility to autoimmune diseases can aid in identifying diagnostic or prognostic biomarkers and the development of personalized therapies. Loci regulating splicing that overlap with genetic risk factors for immune system disorders could serve as valuable starting points for such an approach. There are many steps required to fully understand the complexity of AS and isoform functionality. However, it is clear that there is significant potential in this field.

## Figures and Tables

**Figure 1 ijms-25-08123-f001:**
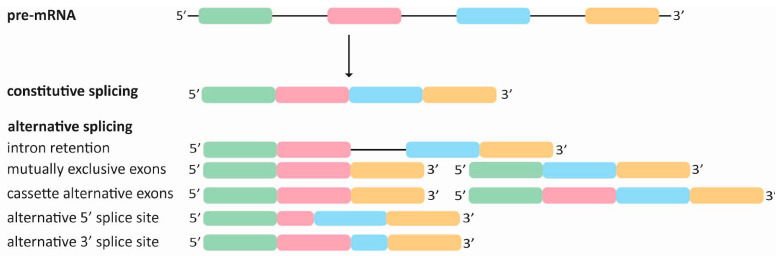
Schematic overview of splicing outcome effects on mRNA. Schematic depiction of a pre-mRNA containing four exons (colored boxes) and three introns (black lines). Constitutive splicing results in a mature transcript that contains all four exons. Different types of alternative splicing can have different outcomes. Intron retention results in the inclusion of one or more introns in the mRNA. With mutually exclusive exons, either one exon or another exon is incorporated but not both exons simultaneously. Cassette alternative exons can either be skipped or incorporated into the mature mRNA. Alternative splicing at alternative 5′ or 3′ splice sites will result in the formation of extended or shortened exons.

**Figure 2 ijms-25-08123-f002:**
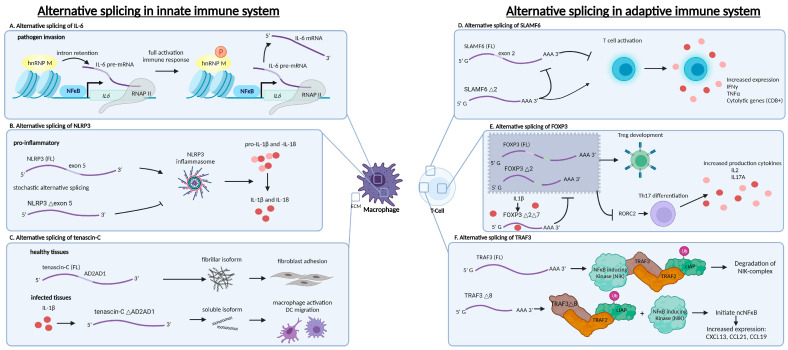
Examples of alternative splicing events that are involved in the innate and adaptive immune response. (**A**) The presence of heterogeneous nuclear ribonucleoproteins (hnRNPs) M on poised chromatin enables immediate interaction with nascent innate immune transcripts, including IL-6. Expression of IL-6 is induced upon the kappa-light-chain-enhancer of B cell (NFκB)-signaling, and the interaction of hnRNP M with the IL-6 pre-mRNAs causes intron retention. This prevents further mRNA processing. Upon full macrophage activation, hnRNP M is phosphorylated and IL-6 transcripts are fully processed and can exit the nucleus. (**B**) In activated macrophages, NLRP3 engages in inflammasome formation and contributes to the cleaving of pro-IL-1β and -IL-18 into functional IL-1β and IL-18. The alternative splicing of NLRP3 results in the generation of short NLRP3 isoforms that lose their pro-inflammatory function. (**C**) IL-1β induces the expression of short isoforms of tenascin-C, which lack the AD2AD1 domain. In contrast to the fibrillar structure of full-length tenascin-C, the short isoform is secreted into the extracellular matrix (ECM) as soluble protein, which activates macrophages and enables dendritic cell (DC) migration. (**D**) SLAMF6 (full length) normally suppresses T cell activation. Absence of exon 2 negates this effect, resulting in T cell activation and subsequent increased expression of cytokines such as IFNγ and TNFα. Additionally, SLAMF6Δ2 inhibits full length SLAMF6. (**E**) FOXP3 (full length) normally inhibits RORC2, thus preventing Th17 differentiation and instead leading T-cells to Treg development. IL-1β induces the alternative splicing of FOXP3 to specifically exclude exon 7 by means of exon skipping. FOXP3(FL) and FOXP3(Δ2) function is inhibited by FOXP3Δ2Δ7, and thus RORC2 is no longer inhibited and T cells will differentiate into Th17 cells with the subsequent production of cytokines IL-2 and IL-17A. (**F**) TRAF3Δ8 is not able to bind NFκB-inducing kinase (NIK), and thus the NIK-complex is not correctly formed and the complex of TRAF3Δ8, TRAF2, and cIAP is degraded. NIK is able to initiate the non-canonical NFκB pathway, which subsequently leads to increased expressions of CXCL13, CCL21, and CCL19.

**Figure 3 ijms-25-08123-f003:**
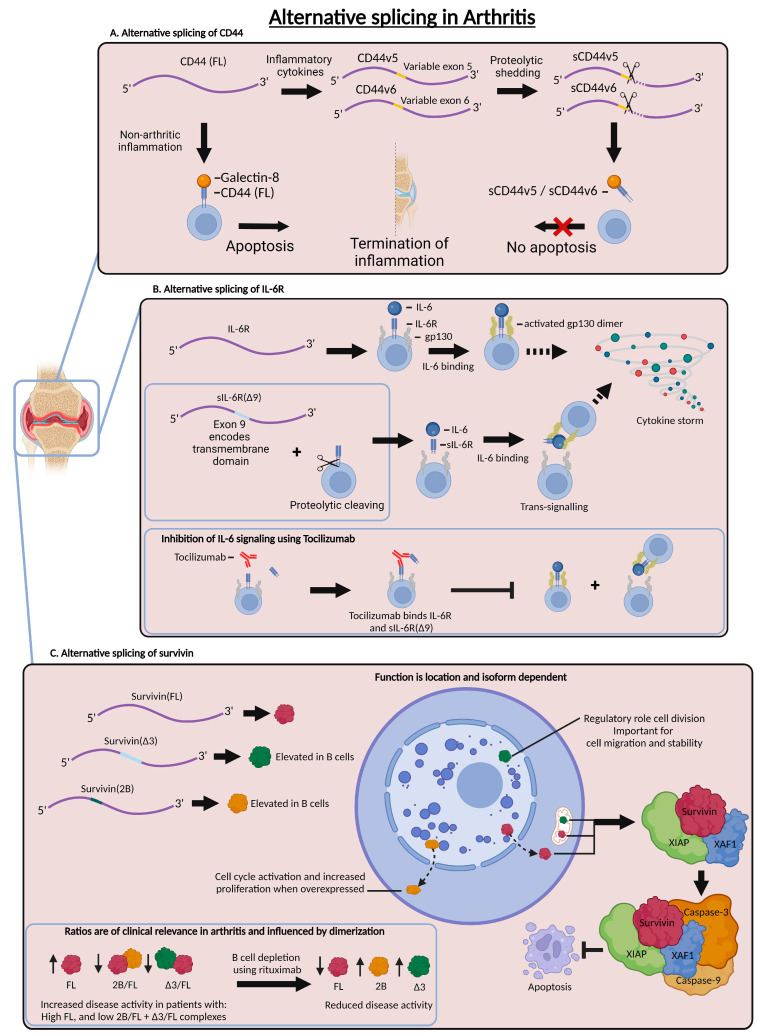
Schematic representation of alternative splicing examples in arthritis. (**A**) Elevated expression levels of CD44v5 and CD44v6 were observed in patients with rheumatoid arthritis. These isoforms were subjected to proteolytic shedding, resulting in soluble isoforms that lack the intracellular signaling domain. Soluble CD44 captures galectin-8 in rheumatoid synovia without subsequently inducing apoptosis, and thus the inflammation will persist. (**B**) In contrast to full-length IL-6R, the delta exon 9 splice variant can be subjected to proteolytic cleaving. Both receptors are overexpressed in patients with arthritis and can induce a cytokine storm upon IL-6 binding and gp130 activation. Tocilizumab binds and inhibits both isoforms of IL-6R and thereby prevents IL-6 signaling. (**C**) Survivin has three different isoforms, FL, Δ3, and 2B. The function of survivin depends on the location and the isoform expressed. Nuclear Δ3 can modulate cell division, migration, and stability. Both FL and Δ3 have anti-apoptotic properties, forming a complex with XIAP and XAF1. This complex binds and inhibits caspase-3 and caspase-9, thereby preventing apoptosis. The 2B can promote cell cycle activation and proliferation. Both Δ3 and 2B are elevated in B cells of arthritis patients, and the ratio between these isoforms influences disease activity. Rituximab-mediated B cell depletion results in reduced FL expression, thereby impairing the expression of FL dimers.

**Figure 4 ijms-25-08123-f004:**
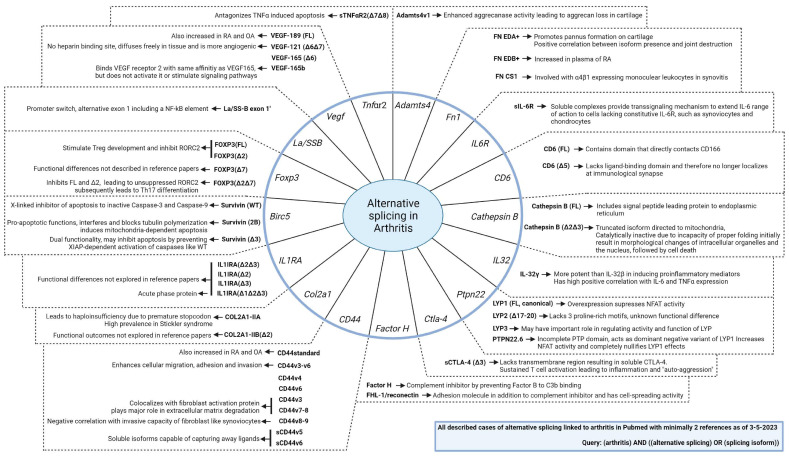
Summarizing overview of alternative splicing events in the scope of arthritis. Known cases of alternative splicing in arthritis are shown in a word web. The gene names are displayed within the outer circle, and corresponding isoforms are described outside of the circle. The splicing isoforms were found using the search query “(arthritis) AND ((alternative splicing) OR (splicing isoform))” on 3rd of May 2023. Isoforms were included when there were at least two references. The references are displayed in a supplementary table and list; respectively, Appendix A.

**Figure 5 ijms-25-08123-f005:**
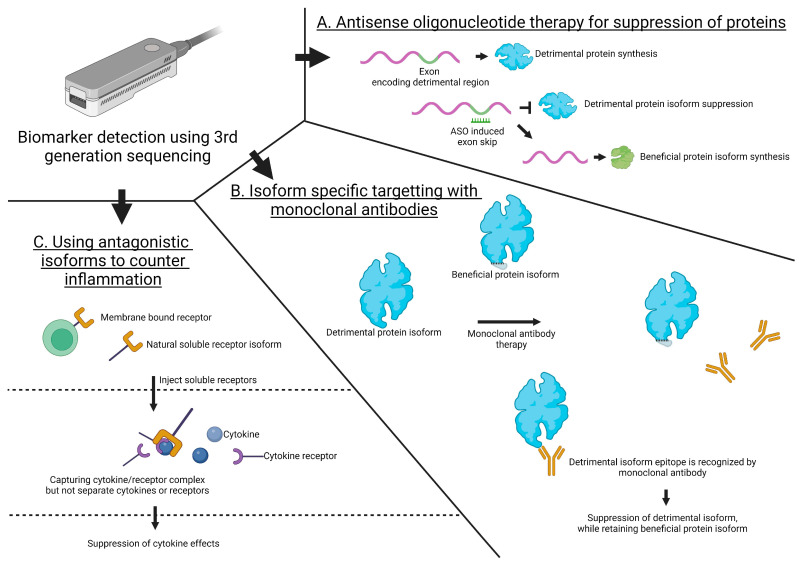
Illustrative examples of potential treatment options based on biomarker detection using 3rd generation sequencing in the scope of arthritis. (**A**) Antisense oligonucleotide (ASO) therapy can be used to induce the specific skipping of a detrimental exon, while maintaining a beneficial protein, depending on the original protein function. (**B**) Monoclonal antibodies capable of recognizing a detrimental protein epitope can be used to specifically target detrimental proteins while retaining beneficial protein isoforms. (**C**) Naturally existing antagonistic isoforms can be used to counter inflammation. Highly purified antagonistic isoforms can be isolated and injected to capture cytokine/receptor complexes, thus suppressing cytokine effects.

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
