# Peer review of "Get Spliced: Uniting Alternative Splicing and Arthritis"

_ijms, 2024, doi:10.3390/ijms25158123_

Round 1

Reviewer 1 Report

Comments and Suggestions for Authors

English writing does not need further correction. Figures and illustrations were well made in the manuscript, which made the clear and vivid clarifications.

However, here are some issues should be concerned as follows:

1. Although there are potential progressive relationships from Section 2 to Section 4, the correlation between Section 1 and Section 2 is relatively weak, probably due to Section 1 plays the role of [Introduction] instead. In this case, an additional [Introduction] part is suggested, with Section 1 being revised by expanding detailed description on Why “Alternative splicing as a post-transcriptional regulator of dynamic cell responses”.

2. Another concern is proposed about the content distribution in each section. Lengthy paragraphs contended in each section, which makes it hard to read without the emphasis on the key points. So, subheadings are suggested.

3. The Review (Reference 31, 2021) cited in this manuscript had a comprehensive summarization on the alternative splicing and its correlation with autoimmune including not only and rheumatoid arthritis, but also systemic lupus erythematosus. So, what’s the innovation point or so-called significance of this Review? Also, referring to the database of PubMed, there have been totally 20 related publishments (including 5 Reviews and 15 Research papers) during 2022 to 2024. However, this article only cited 4 references since 2022 (3 at 2022, 1 at 2023) among the total 58 references. It would be necessary to include as much related novel findings in this Review as possible.

Comments on the Quality of English Language

English should be sharpen to make it more clear and concise to be understood.

Reviewer 2 Report

Comments and Suggestions for Authors

This review explores the relationship between the alternative splicing (AS) process and arthritis, an autoimmune disease. The authors discuss how AS is directly involved in the regulation of immune responses, with dramatic changes in the expression patterns of isoform transcripts in immune cells upon activation. Third-generation RNA sequencing technology has identified thousands of new transcript isoforms, revealing the complexity of AS. Furthermore, genome wide association studies (GWAS) have found that single nucleotide polymorphisms (SNPs) associated with arthritis are enriched in splice sites, suggesting that AS dysregulation may contribute to the pathogenesis of the disease. The paper also examines the therapeutic potential of AS modulation, including the use of isoform-based biomarkers for disease severity or subtype, manipulation of splicing with antisense oligonucleotides, and blockade of specific immune-related isoforms with antibodies.

The review is interesting and the relationship of the splicing process with different pathologies is a topic that many research groups are focusing on due to how little is known about the relationship of alternative splicing with the differences in the phenotype of the patients who suffer from it.

I would like to propose some modifications to improve this review.

- As happens to all of us, there are some typographical errors in the text that should be revised. For example, on line 19 it says "alternative spicing"

- It would be interesting to provide additional details on the aforementioned SNPs obtained by GWAS associated with arthritis.

- Strategies such as antisense oligonucleotides and monoclonal antibodies are mentioned. It would be interesting to include more details on the results of some specific studies.

- I think some more recent bibliography could be included, if possible.

- A very interesting aspect about this review is the therapeutic potential. It would be good to deepen on the this aspect, including a discussion of the challenges and limitations associated with current approaches, as well as future directions for drug development and personalized medicine.

- I think the images should be included with higher quality 
